# PROMPT DISTRIBUTION MATTERS: TUNING VISUAL PROMPT THROUGH SEMANTIC METRIC GUIDANCE

## ABSTRACT

Visual Prompt Tuning (VPT) has become a promising solution for Parameter-Efficient Fine-Tuning (PEFT) of pre-trained Vision Transformer (ViT) models on downstream vision tasks. VPT partially fine-tunes a set of learnable tokens while keeping the majority of the model parameters frozen. Recent research has explored modifying the connection structures of the prompts. However, the fundamental correlation and distribution between the prompts and image tokens remain unexplored. In this paper, we leverage *metric learning* techniques to investigate how the distribution of prompts affects fine-tuning and transfer learning performance. Specifically, we propose a novel framework, **D**istribution **A**ware **V**isual **P**rompt Tuning (DA-VPT), to guide the distributions of the prompts by learning the distance metric from their class-related semantic data. Our method demonstrates that the prompts can serve as an effective bridge to share semantic information between image patches and the class token. We extensively evaluated our approach on popular benchmarks in both recognition and segmentation tasks. The results show the effectiveness of our proposed method and offer a new direction for PEFT optimization in vision transformers. We demonstrate that our approach enables more effective and efficient fine-tuning of ViT models by leveraging semantic information to guide the learning of the prompts, leading to improved performance on various downstream vision tasks. The code will be released.

## 1 INTRODUCTION

With the rapid development of scaling up in the size of models and training datasets (Deng et al., 2009; Sun et al., 2017; Mahajan et al., 2018), a significant number of vision foundation models have been proposed in recent years. In particular, pre-trained models based on the Vision Transformer (ViT) (Dosovitskiy et al., 2020) backbone demonstrate remarkable performance across various computer vision tasks (He et al., 2020; Radford et al., 2021b; He et al., 2022a). Following the powerful abilities of pre-trained vision models, fine-tuning these foundation models for downstream tasks, such as visual recognition (Dosovitskiy et al., 2020) and semantic segmentation (Kirillov et al., 2023), has become a popular strategy.

However, the conventional full fine-tuning strategy, which involves updating all parameters for downstream tasks, is often criticized for its high training costs, overfitting, and the risk of catastrophic forgetting due to mismatches between the scale and distribution of the pre-training and local datasets (Kornblith et al., 2019). These weaknesses particularly affect performance when the scale of the model and data becomes larger (Toneva et al., 2018; Nguyen et al., 2019). To address these challenges, a group of promising research works proposes Parameter-Efficient Fine-Tuning (PEFT), where the majority of model parameters are frozen, and only a small subset of learnable parameters is updated (Kornblith et al., 2019; Houlsby et al., 2019; Pfeiffer et al., 2020; Zaken et al., 2021; Li & Liang, 2021; Chen et al., 2022; Jia et al., 2022).

Starting from the NLP area, Houlsby et al. (2019) and subsequent works (Pfeiffer et al., 2020; Hu et al., 2021) propose updating a minimal number of parameters while achieving performance comparable to full fine-tuning. Extending these techniques to vision tasks, Chen et al. (2022) introduce parallel residual networks alongside the ViT backbone. To further improve the performance of PEFT strategies, Jia et al. (2022) first propose Visual Prompt Tuning (VPT) to prepend a set of learnable tokens, namely *prompts*, to the input data in each ViT layer. This adaptation simplifies

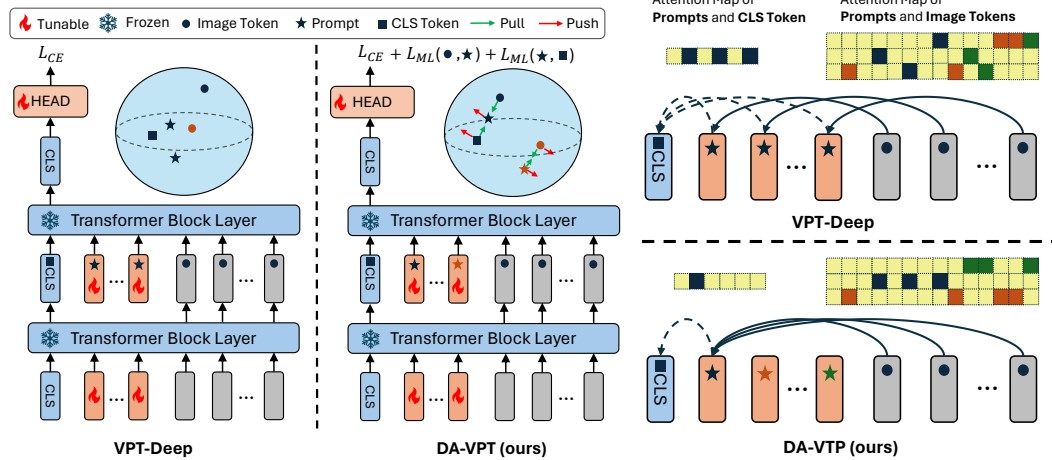

(a) Compare the architecture and feature distributions

(b) Compare how information flow from image patches to CLS token

Figure 1: **Comparing our proposed DA-VPT with the conventional VPT-Deep. VPT-Deep**: ((a) Left and (b) Up) The learnable prompts are fully guided by the single recognition task. Since the distributions of the prompts and image patches are unconstrained, the prompts may attract arbitrary image patches from different classes. This may cause difficulty for the class token to collect the correct information from the samples in its positive class. **DA-VPT**: ((a) Right and (b) Down) The learnable prompts are jointly guided by the main task and the semantic metric learning signal. The distributions of prompts, image patches, and class tokens are based on their semantic clustering. In this case, it is easier for the class token to collect information from the specific prompt with the same class label, simplifying the learning process of the class token.

the transfer learning process by aligning the data distribution of downstream tasks with the original pre-training datasets, showing superior performance even over full fine-tuning strategies. Following the successful VPT, recently Yoo et al. (2023) and Han et al. (2023) propose connecting the prompts across layers with a gating mechanism to dynamically decide the position and number of the prompts.

However, existing VPT approaches mainly focus on manipulating the connection or structure of the prompts, while the intrinsic connection between the prompts and the data representations is often ignored, which causes difficulty in optimizing parameter values in the prompts. The existing VPT work (Jia et al., 2022) and its followers (Yoo et al., 2023; Han et al., 2023; Pei et al., 2024) propose to randomly initialize the prompts, then update them from a single objective of the downstream tasks. We still lack a clear understanding of how the prompts support the model in transfer learning. In this paper, we investigate the correlation between the prompts and the pre-trained model by answering an interesting question: **Can the prompts be guided to deliver information between the image and the class tokens to improve the representation capability?**

To this end, we introduce a novel framework that guides the VPT optimization process by leveraging the semantic connection between the learnable prompts, the visual image patches, and the class tokens. Specifically, we propose connecting the prompts and the visual data by constructing and learning a semantic metric between them in the deep layers of the ViT. For every prompt in the deep layer, we establish a semantic mapping between the prompt and its closest image class. As illustrated in Figure 1 (a), we construct a semantic metric in the latent space by comparing the prompt with the corresponding image patches of the same or different class. Additionally, we construct a semantic metric from the class token to the prompts.

Our key intuition is to improve the similarity between the prompt and the image patches labeled with the same semantic classes while reducing the similarity of those with different classes. By constructing the semantic metric in the feature and prompt space, we demonstrate that semantic information can be easily transferred from image patches to the class tokens through the related prompt in the corresponding class. In essence, our framework employs the related prompt as a *bridge* to connect the class tokens and the semantic information of image patches by effectively guiding their attention maps, as depicted in Figure 1 (b).

Our extensive experiments on 24 popular visual recognition tasks in both Fine-Grained Visual Classification (FGVC) (Jia et al., 2022) and Visual Task Adaptation Benchmark (VTAB-1k) (Zhai et al., 2019) demonstrate substantial improvements over the standard VPT, highlighting the efficacy of our method on both supervised and self-supervised pre-trained models, such as MAE. We also extensively evaluate our method on the ADE20k segmentation task. We demonstrate that our proposal significantly improves the learning efficiency of the prompts with improved performance in downstream tasks while requiring fewer prompts and learnable parameters compared to both the baseline VPT and the other state-of-the-art PEFT methods.

In summary, the main contributions of this paper are as follows:

- We propose **D**istribution **A**ware **V**isual **P**rompt **T**uning (DA-VPT), a novel and efficient framework to improve the learning performance of prompts by constructing semantic metrics between the prompts and the corresponding image feature patches in the deep layers of ViT.
- We reveal the importance of guiding the learning process of the prompts and demonstrate that the prompts can be an effective bridge to connect the semantic information between image patches and class tokens via the attention mechanism.
- We demonstrate the effectiveness of our method on 24 popular downstream visual recognition tasks and 2 segmentation tasks showing significant improvement compared to the vanilla VPT on both supervised and self-supervised pre-trained vision models.

## 2 METHODOLOGY

### 2.1 PRELIMINARY

**The Vision Transformer (ViT)** (Dosovitskiy et al., 2020) is a fundamental model architecture that applies the original Transformer model (Vaswani et al., 2017b) to computer vision tasks. Given an input image $\mathbf{I}$, ViT divides it into a sequence of $N$ flattened 2D patches, which are then linearly projected into a $D$-dimensional embedding space. A learnable [CLS] (Class) token $x_{cls} \in \mathbb{R}^D$ is prepended to the patch embeddings, serving as a global representation for classification tasks. The resulting sequence of embeddings $\mathbf{x} \in \mathbb{R}^{(N+1) \times D}$ is then passed through $L$ Transformer block layers, each consisting of a Multi-Head Self-Attention (MHSA) mechanism which is defined as $\text{MHSA}(\mathbf{x}^l) = \text{Concat}(\text{H}_1, \cdots, \text{H}_h)$ for layer $l \in L$, where each head $H$ computes a scaled dot-product attention $\text{softmax}(\frac{QK^T}{\sqrt{d}}V)$ with subspaces of Query $(Q)$, Key $(K)$, and Value $(V)$ vectors projected from input embedding $x^{l-1}$ in the previous layer. The final output of the ViT is the [CLS] token $x_{cls}^L$, which is used for downstream classification tasks.

**Visual Prompt Tuning (VPT)** (Jia et al., 2022) present a promising PEFT technique for ViT that adapts the pre-trained model to downstream tasks by introducing a small set of learnable parameters, namely prompts. In a specific ViT block layer, a sequence of $M$ learnable prompt tokens $\mathbf{p} = \{p_1, \ldots, p_M\} \in \mathbb{R}^{M \times D}$ is concatenated with the patch embeddings $\mathbf{x} = \{x_1, \ldots, x_N\} \in \mathbb{R}^{N \times D}$. Jia et al. (2022) propose two VPT settings: **VPT-Shallow** where the prompts are only inserted into the first ViT layer, and **VPT-Deep** where the prompts are appended into every ViT layer. We follow the **VPT-Deep** setting since it has a higher capacity and aligns with our proposed method, where the metric learning objective guides the prompts in the deep layer. The resulting sequence of embeddings $[x_{cls}, \mathbf{p}, \mathbf{x}] \in \mathbb{R}^{(M+N+1) \times D}$ is then processed by the next ViT encoder layers. Specifically for image-embedded patches $\mathbf{x}^l$ in the $l$th layer, the output of the $l + 1$ layer can be described as follows:

$$[x_{cls}^{l+1}, [\quad], x_1^{l+1} \ldots x_N^{l+1}] = BLK([x_{cls}^l, p_1^l \ldots p_M^l, x_1^l \ldots x_N^l]), \tag{1}$$

where $p_1^l \ldots p_M^l$ are the $M$ prompts in the $l$th layer, $BLK$ represents the transformer block described above, and the $[\quad]$ represents the empty position which is left for the prompts in the next layer. During fine-tuning, only the visual prompts $\mathbf{p}$ and the linear classification head are updated, while the pre-trained ViT parameters remain frozen.

**Metric Learning (ML)** aims to learn a distance metric that captures the semantic similarity between data points. The *Neighborhood Component Analysis* (NCA) (Roweis et al., 2004) is a popular objective function for metric learning that encourages the learned embeddings to have a higher probability of being correctly classified by the nearest neighbor classifier. Given a set of $N$ labeled

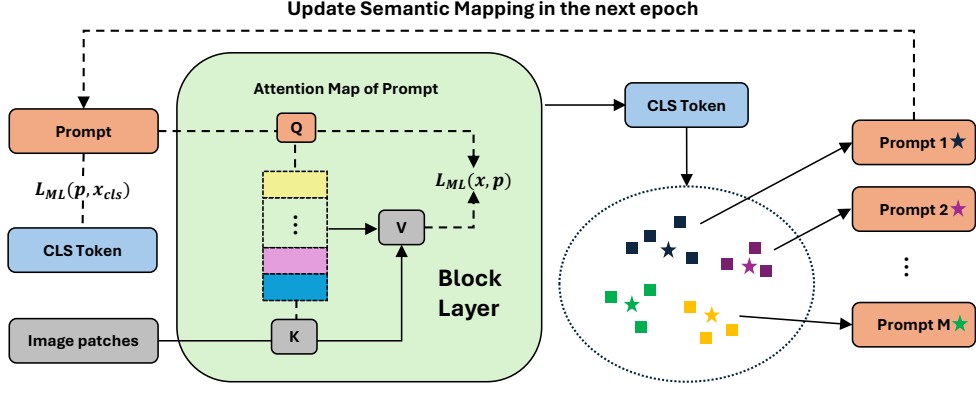

Figure 2: **Distributed Award Visual Prompt Tuning with Semantic Mapping.** This figure illustrates the framework of our proposed method. We establish a semantic mapping between prompts and image classes by clustering class representations into $M$ clusters, where $M$ is the number of prompts. The prompts are guided by a metric space using the smoothed proxy NCA loss $\mathcal{L}_{\text{ML}}$ between prompts and output image tokens based on attention maps. This enables each prompt to attract information from a subset of classes according to its assigned semantic cluster. A similar metric space is established between the [CLS] token and prompts. The semantic mapping is iteratively updated after each training epoch, effectively distributing the learning of visual prompts across the semantic space, enabling them to capture fine-grained class-specific information and improve downstream visual recognition performance.

data points $(x_i, y_i)_{i=1}^N$, where $x_i \in \mathbb{R}^D$ is the input feature vector and $y_i \in \mathbb{R}^C$ is the corresponding class label, the NCA objective is defined as:

$$\mathcal{L}_{\text{NCA}} = -\sum_{i=1}^N \log \frac{\sum_{j \in \mathcal{N}_i} \exp(-D(\mathbf{x}_i, \mathbf{x}_j)/\tau)}{\sum_{k \neq i} \exp(-D(\mathbf{x}_i, \mathbf{x}_k)/\tau)}, \tag{2}$$

where $\mathcal{N}_i = \{j \mid y_j = y_i, j \neq i\}$ denotes the set of neighboring points with the same class. $D(\cdot, \cdot)$ represents a typical *Mahalanobis distance* metric, which in our case is defined as the *cosine similarity*: $D(x_i, x_j) = \hat{x}_i \cdot \hat{x}_j$ where $\hat{x} = \frac{x}{\|x\|_2}$ represents the L2-normalization of vector $x$. By minimizing the NCA loss, the data points are encouraged to align with other data points from the same class while being pushed apart from other classes. Following the NCA loss, later works (Teh et al., 2020; Kim et al., 2020) set up a set of learnable representations $\mathbf{p} = \{p_i \in \mathbb{R}^d\}_{i=1}^C$, named *proxies*, to represent the $C$ classes of the data and compare with other samples as the anchors. In this paper, we propose to set the prompt in the deep layer as the proxy of a subset of classes where the semantic meanings are close.

## 2.2 METRIC LEARNING ON THE LEARNABLE PROMPTS

Our objective is to establish a metric in the feature space that quantifies the distance between learnable prompts and either image patches or the [CLS] token. We hypothesize that, within each layer, a specific prompt should selectively capture information from a subset of relevant classes rather than searching indiscriminately across the entire class space. This approach aims to make the prompts more discriminative and optimize the [CLS] token to collect task-specific information from each class.

For a Transformer block $BLK_l$ at a deep layer $l$ ($l > 0$), we regularize the learning of the prompts $\mathbf{p}^l$ by constructing a space metric between the normalized prompts $\hat{p}^l_i$ and the normalized image patch embeddings $\hat{x}^l_i$. For each prompt $\hat{p}^l_k \in \mathbf{p}^l$ with assigned class label $k$, we aim to satisfy the following constraint for image patch samples $\hat{x}^l_i$ and $\hat{x}^l_j$ in the same batch but with different class labels $i$ and $j$:

$$\hat{p}^l_k \cdot \hat{x}^l_i - \delta \geq \hat{p}^l_k \cdot \hat{x}^l_j + \delta \quad \forall i, j, k, y_k = y_i \neq y_j, \tag{3}$$

where $\cdot$ denotes the dot product, $y_i$ and $y_j$ are the class labels of image patches $x^l_i$ and $x^l_j$, respectively, and $\delta$ is the margin. The intuition is that we want the cosine similarity between the prompt and

patches labeled in the same class to be greater than those labeled in different classes. Since the cosine similarity naturally aligns the comparison of attention maps between the Query and the Key vector, we argue that the $(p_k, x_i)$ pair which is closer in the unique spherical space (measured by cosine similarity) would have a higher chance of being matched in the optimized attention map.

To efficiently build a metric space that satisfies this constraint, we adopt the ML loss proposed by Kim et al. (2020) to compare the learnable prompts with image patches using the smoothed NCA loss (or named Proxy-Anchor loss). Thus, our metric guidance objective between the image patches $\mathbf{x}^l$ and prompts $\mathbf{p}^l$ can be shown as follows:

$$\mathcal{L}_{\text{ML}}(\mathbf{x}, \mathbf{p}) = \frac{1}{|P^+|} \sum_{p_k \in P^+} \left[ \text{LSE}_0^+ \left( - \left( \hat{p}_k \cdot \hat{x}_i - \delta \right) / \tau \right) \right] + \frac{1}{|P|} \sum_{p_k \in P} \left[ \text{LSE}\,0^+ \left( \left( \hat{p}_k \cdot \hat{x}_j + \delta \right) / \tau \right) \right], \tag{4}$$

where $\text{LSE}\,0^+(x) = \log(1 + \sum_{i=1}^{N} e^{x_i})$ is the smoothed LogSumExp with the first argument set to 1, $X_p^+$ denotes the set of image patches with the same label as prompt $p$, $X_p^-$ is its complement set, $\tau$ is the temperature, and $\delta$ is the margin. This objective pushes the prompt $\hat{p}_k^l$ towards the image patches in the positive set $X_p^+$ while pulling it away from the image patches in the negative set $X_p^-$. Practically, we found that comparing the projected Query vector of the prompt $p_Q = p^l \cdot W_Q^l$ has higher performance.

Consequently, we also propose a similar loss $\mathcal{L}_{\text{ML}}(\mathbf{p}, \mathbf{x}_{\text{cls}})$ that aims to pull the [CLS] token closer to the corresponding prompt and push it away from the prompts of different classes. Thus, the overall loss with our metric guidance term can be described as follows:

$$\mathcal{L} = \mathcal{L}_{\text{CE}} + \beta \mathcal{L}_{\text{ML}}(\mathbf{x}, \mathbf{p}) + \lambda \mathcal{L}_{\text{ML}}(\mathbf{p}, \mathbf{x}_{\text{cls}}), \tag{5}$$

where $\beta$ and $\lambda$ are hyperparameters. By jointly optimizing $\mathcal{L}_{\text{ML}}(\mathbf{x}, \mathbf{p})$ and $\mathcal{L}_{\text{ML}}(\mathbf{p}, \mathbf{x}_{\text{cls}})$, our method encourages the prompts to capture class-specific information and aligns the [CLS] token with the relevant prompts, leading to improved fine-tuning performance.

## 2.3 Projection and Saliency Patch Selection

To ensure that the selected prompts effectively focus on critical information in an image sample and filter out false positive image patches, we propose selecting saliency information from the image patches as positive and negative samples for comparison with the prompts in the loss function $\mathcal{L}_{\text{ML}}(\mathbf{x}, \mathbf{p})$. A straightforward approach is to extract the saliency patches from the attention map queried by the prompts. However, in practice, generating the attention map can be resource-intensive, particularly under optimized attention mechanisms such as Flash Attention. To mitigate this, we propose an alternative approach: instead of relying on the attention map, we use the output representation immediately following the attention layer to compare with the prompts. As shown in Figure 2, the output representation $\mathbf{x}^l = \text{MHSA}(x^l)$ is a concatenation of the representations from each head, which also serves as a saliency aggregation of the image patches.

## 2.4 Dynamically Mapping Classes and Prompts

We set $M$ learnable prompts in each layer of the ViT block, where $M \ll C$ (total number of classes). This is to avoid optimization difficulties and unequal training opportunities. Thus, we develop a semantic mapping strategy to map $C$ classes to the $M$ prompts. Before training, we run an additional epoch to pick samples and perform mean pooling to [CLS] token for each class to obtain the set of class representations $\mathbf{s} \in \mathbb{R}^{C \times D}$. We generate these class representations by running the samples through the pre-trained ViT. We then use k-means clustering to group these representations into $M$ clusters, assigning classes to prompts based on the clusters, as illustrated in Figure 2.

To maintain the semantic mapping, we update the dynamic mapping after each epoch. During the training of each epoch, we collect and calculate the mean of class representations $\mathbf{s}$. We then update the k-means with the centroids from the previous epoch and adjust the class-prompt mapping accordingly.

## 2.5 Efficient Bias Tuning

To further improve the learning capability of the image patches in the deep layers, we investigate the partial release of the bias terms in the ViT backbone, as suggested by Zaken et al. (2021). Interestingly,

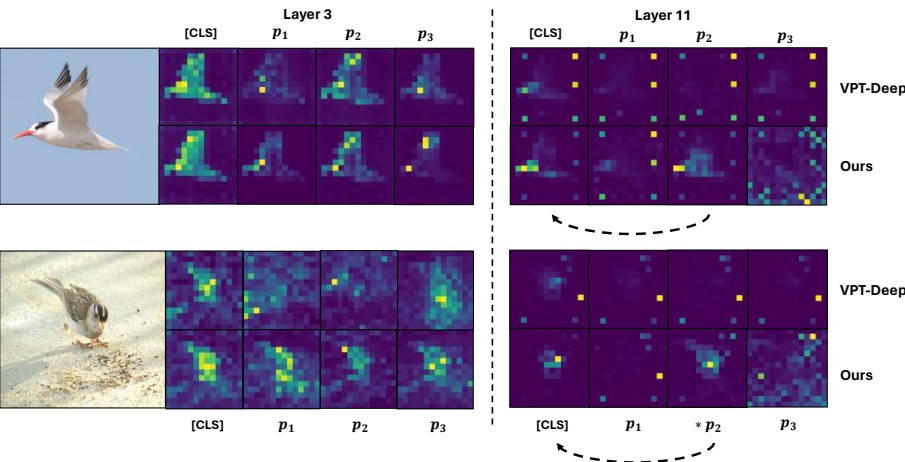

Figure 3: **Visualizing Attention Maps:** We compare the attention maps between VPT-Deep and our proposed method using samples from CUB in both the shallow layer (**Left**, layer 3) and the deep layer (**Right**, layer 11) as examples. For each layer, we illustrate the attention maps of the CLS token and sampled prompts. The $*p$ in layer 11 represents the prompt that is guided as the positive prompt to the CLS token, while the others in this layer are negative prompts. We also show more visualization examples in Appendix.

we found that the fine-tuning performance can be significantly improved when we partially enable the bias terms with our proposed metric guidance loss. We found that the most efficient part is the bias of the linear projection of K and V in the self-attention mechanism (as illustrated in Figure 4 **Left**). This observation is consistent with the findings of existing work (Zaken et al., 2021; Cordonnier et al., 2020). By partially allowing the bias terms to adapt to the downstream task, we provide the model with additional flexibility to adjust the distributions of the image patches and better capture task-specific information under our metric guidance loss.

## 3 TECHNIQUE DISCUSSION

### 3.1 INSIGHTS INTO SIMILARITY AND ATTENTION

In this section, we explore the relationship between attention values and token similarity by analyzing how small changes in the similarity between a prompt and an image patch affect attention weights through gradient updates. Specifically, for a prompt $p$ and an image patch $v_i$, we examine the effect when the attention weight $a_i$ is updated by gradient. Let $\Delta p$ represent a small perturbation that brings $p$ closer to $v_i$, and we introduce the following theorem:

**Theorem 1.** *For a weight perturbation $\Delta a_i$ calculated using the softmax function, there is an approximate relationship:*

$$\Delta a_i \approx a_i(1 - a_i)\Delta s_i,$$

*where $\Delta s_i$ is a small change in the attention score $s_i$, and*

$$\Delta s_i = \frac{\Delta p^\top v_i}{\sqrt{d}}.$$

Using this approximation, we show that a positive gradient change in the attention weight $a_i$ occurs when $\Delta a_i \approx a_i(1 - a_i)\Delta s_i > 0$, which happens when $\Delta s_i = \frac{\Delta p^\top v_i}{\sqrt{d}} > 0$, meaning $p$ moves closer to $v_i$. Conversely, if $p$ moves farther from $v_i$, $\Delta a_i$ decreases. The proof and more details are listed in the Appendix.

### 3.2 VISUALIZE THE GUIDED ATTENTION MAP

To investigate the effect of our proposed metric guidance loss on fine-tuning, we visualize attention maps between image patches and prompts in both shallow and deep layers (Figure 3). In the shallow layer (Figure 3 **Left**), prompts from both VPT-Deep and our method focus on different object subareas,

but our method shows better variety and precision in capturing relevant information. In the deep layer (Figure 3 **Right**), attention maps become sparser as token information becomes more abstract. In particular, VPT-Deep prompts select little valuable information compared to the [CLS] token, whereas our positive prompt ($*p$) successfully identifies informative patches, which are further selected by the [CLS] token. This shows that our method enables better information flow from the positive prompt to the [CLS] token, enhancing the model's ability to capture discriminative features. The visualization of attention maps demonstrate that the positively labeled prompt can serve as an efficient "bridge" to help the [CLS] token collect semantic information in the deep layer, enhancing its discriminative power and facilitating more effective fine-tuning of the ViT backbone.

It is important to note that the information flow from image patches may also include artifacts, as reported in recent work (Darcet et al., 2024). We observed these artifacts in the attention maps of the prompts, as shown in Figure 3. This occurs because both VPT and our method introduce prompts only during the fine-tuning phase, while Darcet et al. (2024) proposed adding them during training from scratch. As a result, the training process in our method is significantly shorter than in their work. However, by comparing the attention maps of the original VPT and our proposed method, we still find that our guiding approach slightly reduces the number and distribution of artifacts, keeping them more contained within the regions of semantic objects. These artifacts warrant further investigation to understand their underlying causes and potential impact on model performance, offering an interesting direction for future research.

## 4 EXPERIMENTS

### 4.1 DATASETS AND SETTINGS

We evaluated our proposed method on two widely-used visual transfer learning benchmarks: Fine-Grained Visual Classification (**FGVC**) (Jia et al., 2022) that contains 5 datasets for general visual recognition, and the Visual Task Adaptation Benchmark (**VTAB-1K**) (Zhai et al., 2019) with 19 datasets for few-shot transfer learning. Additionally, we tested our approach on **ADE20K** (Zhou et al., 2019) and **PASCAL Context** (Mottaghi et al., 2014) for dense prediction tasks.

We use the plain Vision Transformer (ViT) (Dosovitskiy et al., 2020) as the pretrained backbone. To evaluate the generalization of our method, we initialize the backbone using either supervised pre-training on ImageNet-21K (Deng et al., 2009) or self-supervised pre-training on ImageNet-1K without labels, using methods such as MoCo v3 (Chen et al., 2021) and MAE (He et al., 2022a). For architecture, we adopt the base-size model **ViT-B** (12 layers) for visual classification tasks and the large-size model **ViT-L** (24 layers) for semantic segmentation, consistent with existing works. For more experimental details and data set settings, see Section B and Table 5 in the Appendix.

For all of our experiments, we follow two branches of settings: **DA-VPT**, our principal proposed method, which builds on the conventional VPT-Deep Jia et al. (2022) architecture while incorporating our proposed metric learning losses $\mathcal{L}_{\mathrm{ML}}(\mathbf{x}, \mathbf{p})$ and $\mathcal{L}_{\mathrm{ML}}(\mathbf{p}, \mathbf{x}_{\mathrm{cls}})$, as introduced in Section 2.2. **DA-VPT+** is an advanced version of DA-VPT, where we further apply efficient bias tuning to enhance learning capability, as detailed in Section 2.5.

### 4.2 RESULT COMPARISON WITH RECENT STATE-OF-THE-ART

As shown in Table 1, our proposed method, DA-VPT+, consistently outperforms previous methods, including VPT-Deep and E2VPT, across both supervised (ViT) and self-supervised (MAE and MoCo-v3) pre-trained models. On the supervised ViT-B backbone, DA-VPT+ surpasses VPT-Deep by 2.83 *percentage points (pp)* on FGVC and 4.18 *pp* on VTAB-1K, while outperforming E2VPT by 2.72 *pp* on FGVC and 2.20 *pp* on VTAB-1K. For the self-supervised backbones, DA-VPT+ demonstrates even more significant improvements over VPT-Deep and other related works. Remarkably, our method also outperforms the full fine-tuning baseline on average across all three pre-trained backbones while using significantly fewer tunable parameters. The improvements range from 3.40 *pp* to 5.34 *pp* on FGVC tasks and 3.98 *pp* to 7.18 *pp* on VTAB-1K tasks, depending on the pre-trained backbone. These results demonstrate the effectiveness and generalizability of DA-VPT+ across various downstream tasks and pre-trained backbones.

| Methods | Mean Param (M) | FGVC Mean Acc (5) | VTAB-1K | | | |
|---|---|---|---|---|---|---|
| | | | Natural (7) | Specialized (4) | Structured (8) | Mean Acc |
| ViT-B with Supervised pretrained on ImageNet-21k | | | | | | |
| Full | 85.98 | 88.54 | 75.88 | 83.36 | 47.64 | 68.96 |
| VPT-Shallow | 0.11 | 84.62 | 76.81 | 79.68 | 46.98 | 67.82 |
| VPT-Deep | 0.64 | 89.11 | 78.48 | 82.43 | 54.98 | 71.96 |
| E2VPT (Han et al., 2023) | 0.33 | 89.22 | 80.01 | 84.43 | 57.39 | 73.94 |
| DA-VPT (ours) | 0.21 | 91.22 | 80.25 | 85.12 | 58.71 | 74.69 |
| DA-VPT+ (ours) | 0.24 | **91.94** | **81.98** | **86.47** | **59.96** | **76.14** |
| ViT-B with MAE pretrained on ImageNet-1K | | | | | | |
| Full | 85.8 | 82.80 | 59.31 | 79.68 | 53.82 | 64.27 |
| VPT-Shallow | 0.10 | 57.84 | 39.96 | 69.65 | 27.50 | 45.70 |
| VPT-Deep | 0.20 | 72.02 | 36.02 | 60.61 | 26.57 | 41.73 |
| GateVPT (Yoo et al., 2023) | 0.17 | 73.39 | 47.61 | 76.86 | 36.80 | 53.09 |
| E2VPT (Han et al., 2023) | 0.06 | – | 59.52 | 77.80 | 44.65 | 60.66 |
| DA-VPT (ours) | 0.20 | 82.17 | 62.14 | 79.14 | 54.31 | 65.19 |
| DA-VPT+ (ours) | 0.22 | **83.20** | **66.59** | **82.96** | **59.28** | **69.61** |
| ViT-B with MoCo-V3 pretrained on ImageNet-1K | | | | | | |
| Full | 85.8 | 84.25 | 71.95 | 84.72 | 51.98 | 69.55 |
| VPT-Shallow | 0.11 | 79.26 | 67.34 | 82.26 | 37.55 | 62.38 |
| VPT-Deep | 0.20 | 83.12 | 70.27 | 83.04 | 42.38 | 65.90 |
| GateVPT (Yoo et al., 2023) | 0.17 | 83.00 | 74.84 | 83.38 | 49.10 | 69.11 |
| E2VPT (Han et al., 2023) | 0.11 | – | 76.47 | **87.28** | 54.91 | 72.88 |
| DA-VPT (ours) | 0.21 | 85.02 | 74.24 | 83.21 | 55.23 | 70.90 |
| DA-VPT+ (ours) | 0.24 | **86.16** | **76.86** | 84.71 | **58.98** | **73.53** |

Table 1: **Comparison of fine-tuning methods under different pre-trained backbones.** We evaluate our DA-VPT, previous related works and baseline methods on all 24 vision tasks (5 FGVC and 19 VTAB-1K benchmarks) using three types of pre-trained models: Supervised ViT, self-supervised MAE (He et al., 2022a), and self-supervised MoCo-v3 (Chen et al., 2021). We show the mean value of the tasks on FGVC and VTAB-1k. Results are averaged over three trials with different seeds. Top-1 accuracy (%) is reported and the best result is in bold. Detailed results for each task in the VTAB-1K benchmark are presented in Table 7 of the Appendix.

| Method | #Param | ADE20K | | PASCAL Context | |
|---|---|---|---|---|---|
| | | mIoU-SS | mIoU-Ms | mIoU-SS | mIoU-Ms |
| Full-Tuning | 317.3M | 47.60 | 49.18 | 53.69 | 55.21 |
| Linear | 13.1M | 38.09 | 39.16 | 46.06 | 48.13 |
| Bias | 13.2M | 43.61 | 45.73 | 45.15 | 46.47 |
| VPT (baseline) | 13.6M | 44.08 | 46.01 | 49.51 | 51.13 |
| SPT-LoRA (He et al., 2023) | 14.6M | 45.40 | 47.50 | – | – |
| SPT-Adapter (He et al., 2023) | 14.6M | 45.20 | 47.20 | – | – |
| DA-VPT (ours) | 13.6M | 45.10 | 47.07 | 50.50 | 52.37 |
| DA-VPT+ (ours) | 13.7M | **46.47** | **47.61** | **52.52** | **54.58** |

Table 2: **Results of Semantic Segmentation on ADE20K and PASCAL Context.** We report mIoU-SS (single-scale inference) and mIoU-MS (multi-scale inference). All experiments use the **ViT-L** backbone pre-trained on ImageNet-21K. The #Param column indicates the total number of tunable parameters in the entire framework. For SPT (He et al., 2023), we report the results from the original paper, while for other settings and our baseline, we provide our reproduced results. We highlight the best results other than the full fine-tuning.

Table 2 demonstrate our proposed methods, DA-VPT and DA-VPT+, achieve significant improvements over existing baselines and competitive methods in semantic segmentation tasks on both the ADE20K and PASCAL Context datasets. Compared to classification tasks, dense prediction tasks such as segmentation are much more challenging. Notably, lightweight PEFT methods like Linear or Bias exhibit low efficiency compared to full fine-tuning. In such challenging tasks, our proposed DA-VPT+ still achieves comparable performance while using only 4.3% of the tunable parameters, demonstrating both high parameter efficiency and effectiveness across both datasets.

Table 3 compares various state-of-the-art PEFT methods on the FGVC (Jia et al., 2022) using the ViT-B model pre-trained on ImageNet-21K. Our proposed DA-VPT and DA-VPT+ methods demonstrate superior performance across FGVC datasets. DA-VPT+ achieves the highest mean accuracy of

| Method \ Dataset | CUB-200 -2011 | NABirds | Oxford Flowers | Stanford Dogs | Stanford Cars | Mean Acc (%) | Mean Params (M) |
|---|---|---|---|---|---|---|---|
| Full fine-tuning (Jia et al., 2022) | 87.3 | 82.7 | 98.8 | 89.4 | 84.5 | 88.54 | 85.98 |
| Linear Probing (Jia et al., 2022) | 85.3 | 75.9 | 97.9 | 86.2 | 51.3 | 79.32 | **0.18** |
| Adapter (Houlsby et al., 2019) | 87.1 | 84.3 | 98.5 | 89.8 | 68.6 | 85.67 | 0.41 |
| Bias (Zaken et al., 2021) | 88.4 | 84.2 | 98.8 | 91.2 | 79.4 | 88.41 | 0.28 |
| AdaptFormer (Chen et al., 2022) | 87.4 | 84.8 | 99.0 | 90.7 | 81.0 | 88.58 | 1.54 |
| VPT-Shallow (Jia et al., 2022) | 86.7 | 78.8 | 98.4 | 90.7 | 68.7 | 84.62 | 0.25 |
| VPT-Deep (Jia et al., 2022) | 88.5 | 84.2 | 99.0 | 90.2 | 83.6 | 89.11 | 0.85 |
| SSF (Lian et al., 2022) | 89.5 | 85.7 | 99.6 | 89.6 | 89.2 | 90.72 | 0.39 |
| SNF (Wang et al., 2023) | 90.2 | 87.4 | 99.7 | 89.5 | 86.9 | 90.74 | 0.25 |
| MP(Gao et al., 2023b) | 89.3 | 84.9 | 99.6 | 89.5 | 83.6 | 89.38 | 1.20 |
| E2VPT (Han et al., 2023) | 89.1 | 84.6 | 99.1 | 90.5 | 82.8 | 89.22 | 0.65 |
| MoSA (Zhang et al., 2024) | 89.3 | 85.7 | 99.2 | **91.9** | 83.4 | 89.90 | 1.54 |
| VPT (Baseline) | 88.6 | 85.7 | 99.2 | 89.0 | 87.4 | 90.14 | 0.36 |
| DA-VPT (Ours) | 90.2 | 87.4 | 99.4 | 89.4 | 89.7 | 91.22 | 0.30 |
| DA-VPT+ (Ours) | **90.8** | **88.3** | **99.8** | 89.8 | **91.0** | **91.94** | 0.32 |

Table 3: **Comparison of various fine-tuning methods on different downstream tasks.** The ViT-B model pre-trained on ImageNet-21K is used as basic backbone. Results are averaged over three trials with different seeds. Top-1 accuracy (%) is reported and the best result is in **bold**.

Table 4: **Ablation study on different components in our DA-VPT on two datasets: CUB-200-2011 in FGVC and *Natural* in VTAB-1k.** For each $\mathcal{L}_{\mathrm{ML}}$ component, we also search for its optimal hyperparameter. The learnable [CLS] token is combined with Efficient Bias for simplicity. The latency and memory are tested in the same server with RTX4090 GPU.

| Components of our Techniques | | | VTAB-1k *Natural (7)* | | FGVC CUB-200 | | Latency | Memory |
|---|---|---|---|---|---|---|---|---|
| $\mathcal{L}_{\mathrm{ML}}(\mathbf{x}, \mathbf{p})$ | $\mathcal{L}_{\mathrm{ML}}(\mathbf{p}, \mathbf{x}_{\mathrm{cls}})$ | Efficient Bias | Param | Accuracy | Param | Accuracy | (ms/img) | (GB) |
| | | | | 79.45 (base) | | 88.64 (base) | 1.41 | 2.41 |
| ✓ | | | 0.14M | 79.47 (+0.02) | 0.20M | 89.24 (+0.60) | 1.51 | 2.41 |
| | ✓ | | (0.16%) | 79.51 (+0.06) | (0.24%) | 89.06 (+0.42) | 1.52 | 2.41 |
| ✓ | ✓ | | | 80.53 (+1.08) | | 89.86 (+1.22) | 1.54 | 2.41 |
| | | ✓ | | 80.06 (+0.61) | | 89.55 (+0.91) | 1.45 | 2.76 |
| | ✓ | ✓ | 0.16M | 81.02 (+1.57) | 0.23M | 90.41 (+1.77) | 1.53 | 2.76 |
| ✓ | | ✓ | (0.19%) | 81.50 (+2.05) | (0.27%) | 90.54 (+1.90) | 1.53 | 2.76 |
| ✓ | ✓ | ✓ | | 81.98 (+2.53) | | 90.89 (+2.25) | 1.56 | 2.76 |

91.94% across all datasets, surpassing previous state-of-the-art methods like SNF (Wang et al., 2023) and MoSA (Zhang et al., 2024). It shows particularly strong performance on the CUB and Cars datasets, where it achieves the highest accuracy, surpassing the previous SOTA by 0.6 and 1.8 *pp* respectively. Notably, DA-VPT+ outperforms full fine-tuning by a significant margin while using only a fraction of the parameters. Both DA-VPT and DA-VPT+ also show consistent improvements over the VPT baseline by 1.80 *pp*, while requiring fewer prompts and parameters. This performance is achieved with parameters comparable to most other PEFT methods, demonstrating an excellent balance between accuracy and parameter efficiency.

## 4.3 ABLATION STUDIES AND DISCUSSION

### 4.3.1 ABLATION STUDY

The ablation study demonstrates the individual and collective contributions of each component in our proposed DA-VPT method on the CUB-200-2011 dataset from the FGVC benchmark and the Natural task category from the VTAB-1k benchmark. The metric learning losses, $\mathcal{L}_{\mathrm{ML}}(\mathbf{x}, \mathbf{p})$ and $\mathcal{L}_{\mathrm{ML}}(\mathbf{p}, \mathbf{x}_{\mathrm{cls}})$, lead to accuracy improvements of 1.08 percentage points (pp) on VTAB-1k Natural and 1.22 pp on CUB-200-2011 over the baseline. The integration of Efficient Bias further enhances the performance, contributing to an additional 1.45 pp and 1.03 pp improvement on the respective datasets when combined with both metric learning losses.

When all three components are combined, our DA-VPT method achieves the highest performance, with total accuracy improvements of 2.53 pp on VTAB-1k Natural and 2.25 pp on CUB-200-2011. While the incorporation of these components introduces a minimal increase in latency (from 1.41 ms/img to 1.56 ms/img) and memory usage, the gained accuracy far outweighs this slight trade-off.

The improvements in accuracy achieved by our DA-VPT method on both datasets, ranging from 0.02 pp to 2.53 pp for individual components and their combinations, demonstrate the effectiveness of our proposed approach in enhancing the fine-tuning performance of ViT models. Notably, the combination of $\mathcal{L}_{\mathrm{ML}}(\mathbf{x}, \mathbf{p})$ and Efficient Bias yields substantial improvements with only a modest increase in parameters (0.02M for both datasets). This highlights the efficiency of our method in achieving significant performance gains with minimal parameter overhead.

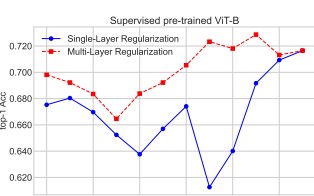 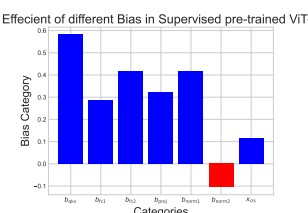 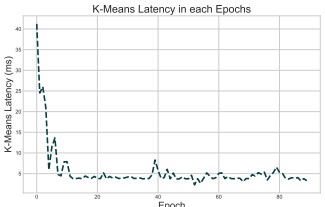

Figure 4: **Left:** Illustrates the impact of the number and position of the layers to which the proposed metric learning loss is applied. **Middle:** This figure shows the latency of the k-means calculation in each epoch. **Right:** Illustrates the importance of each category of efficient bias measured on the CUB-200-2011 dataset.

### 4.3.2 STUDY OF THE GUIDING LAYERS AND OTHER EFFICIENT BIAS

In this section, we investigate which layers and how many prompts can be efficiently guided by our proposed loss. First, we evaluate the impact of applying our metric learning loss to a single layer. As shown by the blue line in Figure 4 (**Left**), applying the loss to the last layer yields the most efficient results in most cases. This is likely because the later layers contain higher-level features compared to earlier layers. We also explore the effect of applying our loss to multiple layers. The red line illustrates the effect of applying the loss from a specific layer to the last layer. We observe that the impact varies significantly across different pre-trained models. Additional results for other models (MAE and MoCo) are provided in the Appendix.

As illustrated in Figure 4 (**Middle**), we demonstrate that certain categories of efficient bias contribute more significantly to the overall performance improvement, highlighting the need for selective optimization of these components. We also evaluate the cost of the progress of the k-means to update the semantic mapping as illustrated in Figure 4 (**Right**). Interestingly, the latency is not a major concern after the first few epochs, as the class representations tend to stabilize over time. This finding suggests that the computational overhead of updating the class-prompt mappings diminishes as training progresses. Impact of other parameters can be referred in Figure 5 of Appendix.

## 5 CONCLUSION

In this paper, we introduce Distribution-Aware Visual Prompt Tuning (DA-VPT), a novel and efficient framework for improving the learning performance of visual prompts in Vision Transformer (ViT) models. By constructing semantic metrics between the prompts and the corresponding image feature patches in the deep layers of ViT, our method effectively guides the learning process of the prompts, enabling them to serve as a bridge connecting the semantic information between image patches and class tokens via the attention mechanism. Through extensive experiments on 24 popular visual recognition tasks and 2 segmentation tasks in different domains, we demonstrated that DA-VPT significantly improves the performance of downstream tasks compared to vanilla VPT, while requiring fewer prompts and learnable parameters. Our results highlight the importance of considering the intrinsic connection between visual prompts and data samples and showcase the potential of our approach to enhance the transfer learning capabilities of pretrained vision models. We believe that our findings can inspire further research on parameter-efficient fine-tuning strategies and contribute to the development of more effective and efficient vision foundation models.

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

APPENDIX:

## A   RELATED WORKS

**Parameter-Efficient Fine-Tuning (PEFT)** Initially introduced by Vaswani et al. (2017a), Transformers have been pre-trained across various domains, including natural language processing (e.g., LLaMA (Touvron et al., 2023), GPT (Brown et al., 2020)) and computer vision (e.g., MAE (He et al., 2022b), CLIP (Radford et al., 2021a), ViT-22b (Dehghani et al., 2023)). Recent advancements in PEFT focus on freezing most parameters while selectively fine-tuning others to improve efficiency for downstream tasks. Kornblith et al. (2019) suggested training only the classification head, whereas Zaken et al. (2021) achieved significant gains by tuning only the bias terms. Further refinements by Lian et al. (2022) and Xie et al. (2023) involve adjusting additional shifting and scaling factors. A subset of PEFT strategies, such as those by Houlsby et al. (2019), involves tuning lightweight models known as adapters alongside Transformer backbones. These approaches were further extended by Pfeiffer et al. (2020) with bidirectional projection networks, and by Chen et al. (2022), who adapted them for computer vision tasks, significantly improving performance. Most recently, Zhang et al. (2024) introduced a combination of sparse adapters, further boosting their efficacy.

**Visual Prompt Tuning (VPT)** This area of research uses learnable vectors, or prompts, to add task-specific information to input data (Li & Liang, 2021; Liu et al., 2023; Lester et al., 2021; Liu et al., 2021). It also uses vision-language models that can be tuned to help with vision tasks (Radford et al., 2021b; Zhou et al., 2022; Ge et al., 2023). Jia et al. (2022) first demonstrated the effective application of prompts in Vision Transformers (ViT), introducing variations like VPT-Shallow at the input layer and VPT-Deep across all Transformer layers. Gao et al. (2022) used visual prompts for test-time domain adaptation, and Gao et al. (2023a) expanded prompt tuning to video recognition. Studies by Tsai et al. (2024) and Ren et al. (2024b) further explored robust visual perception and deep metric learning applications of visual prompts.

Following the seminal work of Jia et al. (2022), later studies such as those by Han et al. (2023) and Yoo et al. (2023) developed dynamic mechanisms for selecting the number and placement of visual prompts. Tu et al. (2023) enhanced VPT by linking intermediate layers directly with task-specific heads, and Pei et al. (2024) introduced a spatial selection mechanism to better coordinate attention between image patches and visual prompts. More recently, Han et al. (2024) analyzed the underlying conditions contributing to the success of VPT. In contrast to existing research that mostly trains visual prompts with a single downstream goal, our study aims to improve prompt effectiveness by revealing the connections and distributions between the prompts and image patches using novel metric learning guidance.

**Metirc Learning (ML)** This field focuses on learning representations and metrics that distinguish the separation between two data samples by arranging similar data points closer together in the representation space. Initial approaches employed *contrastive loss* to differentiate samples from distinct classes (Chopra et al., 2005; Hadsell et al., 2006). Subsequent methods introduced an *anchor* point, utilizing *triplet loss* with a specified margin to compare both positive and negative samples (Weinberger & Saul, 2009; Cheng et al., 2016; Hermans et al., 2017). Further advancements in Metric Learning have leveraged *Neighbourhood Components Analysis (NCA)*, inspired by the semantic relationships between different classes, to explore data distributions and class relationships more deeply (Roweis et al., 2004; Movshovitz-Attias et al., 2017; Teh et al., 2020; Kim et al., 2020; Venkataramanan et al., 2022; Roth et al., 2022). Recent studies have shown the effectiveness and robustness of NCA-based metric learning, particularly when applied to Vision Transformer (ViT) backbones (Ermolov et al., 2022; Patel et al., 2022; Kotovenko et al., 2023). These studies highlight the critical role of data distributions in learning discriminating representations (Wang et al., 2017; Laradji & Babanezhad, 2020; Ren et al., 2019; 2021; 2024a).

In this paper, we extend these principles by examining the distributions and interactions between the prompts and image patches within ViTs. We propose a novel learning framework that utilizes a metric learning objective to guide the configuration and optimization of visual prompts, aiming to enhance their performance in discriminative tasks.

# B    DETAILS ABOUT THE EXPERIMENTS AND REPRODUCIBILITY

**Benchmarks. FGVC** consists of five fine-grained visual classification datasets: CUB-200-2011 (Wah et al., 2011), NABirds (Van Horn et al., 2015), Oxford Flowers (Nilsback & Zisserman, 2008), Stanford Dogs (Khosla et al., 2011), and Stanford Cars (Gebru et al., 2017). Following the protocol outlined in VPT (Jia et al., 2022), we split each dataset into `train` (90%) and `val` (10%) subsets. **VTAB-1K** comprises 19 diverse visual tasks, categorized into three groups: (i) *Natural*, which includes natural images captured by standard cameras; (ii) *Specialized*, consisting of images captured by specialized equipment; and (iii) *Structured*, which involves tasks requiring structural understanding. **ADE20K** is a popular semantic segmentation benchmark containing 150 fine-grained semantic concepts, while **PASCAL Context** provides pixel-wised labels of objects from 60 classes.

**Implementation Details for Classification Tasks** For the FGVC datasets, we apply data augmentation by randomly resizing and cropping the images to a resolution of $224 \times 224$ pixels and applying random horizontal flips. For VTAB-1K, following the protocol in (Zhai et al., 2019; Jia et al., 2022), we directly resize the images to $224 \times 224$ pixels without applying any additional data augmentation techniques. We optimize the model using the *AdamW* optimizer with a mini-batch size of 32 for a total of 100 epochs. We employ a linear warm-up strategy for the first 10 epochs and a cosine learning rate schedule (Loshchilov & Hutter, 2016), which gradually decays the learning rate from its initial value to 1e-8 over the course of training. The initial learning rate and other hyperparameters are determined through cross-validation on the *val* set for each dataset. For all experiments, we report the average accuracy score on the test set over three runs with different random seeds, following the existing works (Jia et al., 2022; Lian et al., 2022; Gao et al., 2023b).

**Implementation Details for Segmentation Tasks**
For segmentation tasks, we adopt the framework from SETR (Zheng et al., 2021) and reproduce the experiments using the MMSegmentation codebase. We use the settings from SETR-PUP, where one primary head and three auxiliary heads are applied to process features from layers 9, 12, 18, and 24. For both ADE20K and PASCAL Context, we follow the original work's protocol, training for 160k and 80k iterations, respectively.

**Dataset Splitting.** We strictly follow the practice of VPT (Jia et al., 2022) to perform the spliting of the train/val/test set. The details of the evaluated tasks and datasets used in the paper can be referred in Tab. 5.

| Datasets | Task Description | Classes | Train Size | Val Size | Test Size |
|---|---|---|---|---|---|
| Fine-Grained Visual Classification (FGVC) (Jia et al., 2022) | | | | | |
| CUB-200-2011 (Wah et al., 2011) | Fine-grained Bird Species Recognition | 200 | 5,394 | 600 | 5,794 |
| NABirds (Van Horn et al., 2015) | Fine-grained Bird Species Recognition | 55 | 21,536 | 2,393 | 24,633 |
| Oxford Flowers (Nilsback & Zisserman, 2008) | Fine-Grained Flower Species recognition | 102 | 1,020 | 1,020 | 6,149 |
| Stanford Dogs (Khosla et al., 2011) | Fine-grained Dog Species Recognition | 120 | 10,800 | 1,200 | 8,580 |
| Stanford Cars (Gebru et al., 2017) | Fine-grained Car Classification | 196 | 7,329 | 815 | 8,041 |
| Visual Task Adaptation Benchmark (VTAB-1k) (Zhai et al., 2019) | | | | | |
| Caltech101 (Fei-Fei et al., 2006) | | 102 | | | 6,084 |
| CIFAR-100 (Krizhevsky et al., 2009) | | 100 | | | 10,000 |
| DTD (Cimpoi et al., 2014) | Natural-Tasks (7) | 47 | | | 1,880 |
| Oxford-Flowers102 (Nilsback & Zisserman, 2006) | Natural images captured using | 102 | 800/1000 | 200 | 6,149 |
| Oxford-PetS (Parkhi et al., 2012) | standard cameras. | 37 | | | 3,669 |
| SVHN (Netzer et al., 2011) | | 10 | | | 26,032 |
| Sun397 (Xiao et al., 2010) | | 397 | | | 21,750 |
| Patch Camelyon (Veeling et al., 2018) | Special-Tasks (4) | 2 | | | 32,768 |
| EuroSAT (Helber et al., 2019) | Images captured via specialized | 10 | 800/1000 | 200 | 5,400 |
| Resisc45 (Cheng et al., 2017) | equipments | 45 | | | 1,880 |
| Retinopathy (Dugas et al., 2015) | | 5 | | | 42,670 |
| Clevr/count (Johnson et al., 2017) | | | | | 15,000 |
| Clevr/distance (Johnson et al., 2017) | | 6 | | | 15,000 |
| DMLab (Beattie et al., 2016) | | 6 | | | 22,735 |
| KITTI-Dist (Geiger et al., 2013) | Structured-Tasks (8) | 4 | 800/1000 | 200 | 711 |
| dSprites/location (Matthey et al., 2017) | Require geometric comprehension | 16 | | | 73,728 |
| dSprites/orientation (Matthey et al., 2017) | | 16 | | | 73,728 |
| SmallNORB/azimuth (LeCun et al., 2004) | | 18 | | | 12,150 |
| SmallNORB/elevation (LeCun et al., 2004) | | 18 | | | 12,150 |
| Image Semantic Segmentation | | | | | |
| ADE20K (Zhou et al., 2019) | Fine-grained images with | 150 | 20210 | 2000 | 3352 |
| PASCAL Context (Mottaghi et al., 2014) | pixel-wise | 60 | 4998 | 5105 | — |
| | semantic annotations | | | | |

Table 5: **The details and specifications of the downstream task datasets we selected to evaluate our proposed framework.**

| Configuration | Value |
|---|---|
| Optimizer | AdamW Loshchilov & Hutter (2017) |
| Base learning rate range | {1e-2, 5e-3, 1e-3, 5e-4, 1e-4, 5e-5} |
| Weight decay range | {0.001, 0.005, 0.01, 0.05, 0.1, 0.5, 1.0} |
| Learning rate schedule | Cosine Decay Loshchilov & Hutter (2016) |
| Batch size | 32 |
| Warmup epoch | 10 |
| Total epoch | 100 (ViT-B/16) |
| Augmentation | RandomResizedCrop, RandomHorizontalFlip |

Table 6: Hyper Parameters Searching Space and Training configuration in our experiments

---

**Algorithm 1** Distribution Aware Visual Prompt Tuning (DA-VPT)

---

**Input:** Pre-trained ViT model $f_\theta$, Dataset $\mathcal{D} = (x_i, y_i)_{i=1}^N$,
number of prompts $M$, $\beta$, $\lambda$, learning rate and other related hyperparameters
**Output:** Fine-tuned ViT model $f_\theta$
Initialize $M$ prompts $\mathbf{p}^l$ for each layer $l$
Get class tokens $\mathbf{S} \in \mathbb{R}^{C \times D}$ by Forward passing $f_\theta$
Create a mapping from $C$ classes to $M$ prompts ($C \to M$) using k-means clustering on $\mathbf{S}$
**while** *stop criteria is not satisfied* **do**
    Obtain a batch $\{x_i, y_i\}_{i=1}^n$ from $\mathcal{D}$
    Forward pass $\mathbf{x}_i$ through ViT $f_\theta$ with prompts $\mathbf{p}^l$
    Select saliency patch $\mathbf{x}$ right after attention layer in last selected blocks
    Calculate metric learning losses $\mathcal{L}_{\text{ML}}(\mathbf{x}, \mathbf{p})$ and $\mathcal{L}_{\text{ML}}(\mathbf{p}, \mathbf{x}_{\text{cls}})$
    Calculate cross-entropy loss $\mathcal{L}_{\text{CE}}$
    Minimize loss: $\mathcal{L} = \mathcal{L}_{\text{CE}} + \beta\mathcal{L}_{\text{ML}}(\mathbf{x}, \mathbf{p}) + \lambda\mathcal{L}_{\text{ML}}(\mathbf{p}, \mathbf{x}_{\text{cls}})$
    Update $\mathbf{p}$ and other learnable parameters from Backward of $\mathcal{L}$
    Update class tokens $\mathbf{S}$ and class-prompt mapping $C \to M$
**return** Fine-tuned ViT model $f_\theta$

---

## C  THE PROOF AND DETIAL OF THEOREM 1

**Theorem 1.** *For a weight perturbation $\Delta a_i$ calculated using the softmax function, there is an approximate relationship:*

$$\Delta a_i \approx a_i(1 - a_i)\Delta s_i,$$

*where $\Delta s_i$ is a small change in the attention score $s_i$, and*

$$\Delta s_i = \frac{\Delta p^\top v_i}{\sqrt{d}}.$$

*Proof.* The attention weights $a_i$ are computed using the softmax function applied to the attention scores $s_i$:

$$a_i = \frac{\exp(s_i)}{\sum_k \exp(s_k)}.$$

Consider a small perturbation $\Delta s_i$ in $s_i$. The corresponding change in $a_i$ can be approximated using a first-order Taylor expansion:

$$\Delta a_i \approx \frac{\partial a_i}{\partial s_i}\Delta s_i.$$

The partial derivative of $a_i$ with respect to $s_i$ is given by:

$$\frac{\partial a_i}{\partial s_i} = a_i(1 - a_i).$$

This follows from differentiating the softmax function:

$$\frac{\partial a_i}{\partial s_j} = a_i(\delta_{ij} - a_j),$$

where $\delta_{ij}$ is the Kronecker delta. When $i = j$, this simplifies to:

$$\frac{\partial a_i}{\partial s_i} = a_i(1 - a_i).$$

Therefore,

$$\Delta a_i \approx a_i(1 - a_i)\Delta s_i.$$

The attention score $s_i$ is defined as:

$$s_i = \frac{p^\top v_i}{\sqrt{d}},$$

where $p$ is the query vector, $v_i$ is the key vector, and $d$ is the dimensionality. A small change $\Delta p$ in $p$ leads to a change in $s_i$:

$$\Delta s_i = \frac{(p + \Delta p)^\top v_i}{\sqrt{d}} - \frac{p^\top v_i}{\sqrt{d}} = \frac{\Delta p^\top v_i}{\sqrt{d}}.$$

Substituting $\Delta s_i$ back into the expression for $\Delta a_i$, we have:

$$\Delta a_i \approx a_i(1 - a_i)\frac{\Delta p^\top v_i}{\sqrt{d}}.$$

This establishes the approximate relationship between the weight perturbation $\Delta a_i$ and the small change in the attention score $\Delta s_i$, as stated in the theorem. □

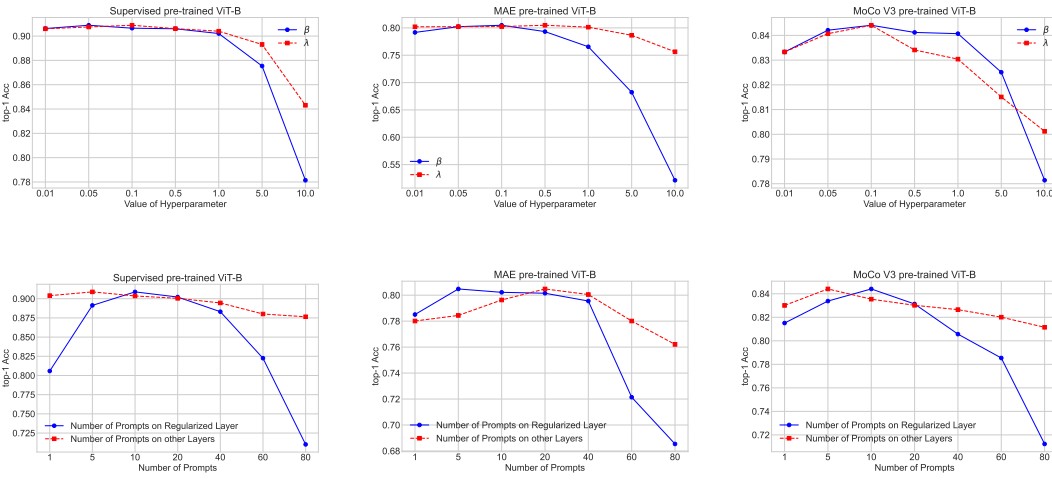

Figure 5: **Impact of Hyperparameters in Three Pre-trained Models on CUB-200-2011**: This figure illustrates the impact of hyperparameters on the performance of our proposed method across three pre-trained models (Supervised ViT, MAE, and MoCo-v3) on the CUB-200-2011 dataset. The hyperparameters investigated include the weight factors $\beta$ and $\lambda$ for the two proposed $\mathcal{L}_{\text{ML}}$ losses, the number of prompts in the metric guidance layer, and the number of prompts in other layers. The results show that the optimal weight factors are less than 1.0, indicating that a balanced contribution from the $\mathcal{L}_{\text{ML}}$ losses is beneficial for performance. Furthermore, the number of prompts in the guidance layer exhibits higher sensitivity compared to the number of prompts in other layers, suggesting that the choice of prompt configuration in the guidance layer plays a crucial role in the effectiveness of our method. These findings provide insights into the importance of carefully tuning the hyperparameters to achieve optimal performance across different pre-trained models.

## D    LIMITATION

As a work that follows the Parameter-Efficient Fine-Tuning (PEFT) paradigm, our method has certain limitations. One challenge is the potential difficulty in searching for the optimal hyperparameters, as our method introduces additional regularization terms, namely the metric learning losses, which work in conjunction with the original cross-entropy loss for the downstream tasks. The effectiveness of these regularization losses is sensitive to the weight ratios $\beta$ and $\lambda$, and their optimal values may vary depending on the backbone model and the specific downstream task. This sensitivity necessitates a careful and potentially time-consuming hyperparameter search to achieve the best performance.

| Methods | Natural (7) | | | | | | | Specialized (4) | | | | Structured (8) | | | | | | | |
|---|---|---|---|---|---|---|---|---|---|---|---|---|---|---|---|---|---|---|---|
| | Caltech101 | CIFAR-100 | DTD | Flowers102 | Pets | SVHN | Sun397 | Patch Camelyon | EuroSAT | Resisc45 | Retinopathy | Clevr/count | Clevr/distance | DMLab | KITTI/distance | dSprites/loc | dSprites/ori | SmallNORB/azi | SmallNORB/ele |
| Full fine-tuning Jia et al. (2022) | 68.9 | 87.7 | 64.3 | 97.2 | 86.9 | 87.4 | 38.8 | 79.7 | 93.7 | 84.2 | 73.9 | 56.3 | 58.6 | 41.7 | 65.5 | 57.5 | 46.7 | 25.7 | 29.1 |
| Linear probing Jia et al. (2022) | 63.4 | 85.0 | 63.2 | 97.0 | 86.3 | 36.6 | 51.0 | 78.5 | 87.5 | 68.6 | 74.0 | 34.3 | 30.6 | 33.2 | 55.4 | 12.5 | 20.0 | 9.6 | 19.2 |
| Adapter Houlsby et al. (2019) | 74.1 | 86.1 | 63.2 | 97.7 | 87.0 | 34.6 | 50.8 | 76.3 | 88.0 | 73.1 | 70.5 | 45.7 | 37.4 | 31.2 | 53.2 | 30.3 | 25.4 | 13.8 | 22.1 |
| Bias Zaken et al. (2021) | 72.8 | 87.0 | 59.2 | 97.5 | 85.3 | 59.9 | 51.4 | 78.7 | 91.6 | 72.9 | 69.8 | 61.5 | 55.6 | 32.4 | 55.9 | 66.6 | 40.0 | 15.7 | 25.1 |
| VPT-Shallow Jia et al. (2022) | 77.7 | 86.9 | 62.6 | 97.5 | 87.3 | 74.5 | 51.2 | 78.2 | 92.0 | 75.6 | 72.9 | 50.5 | 58.6 | 40.5 | 67.1 | 68.7 | 36.1 | 20.2 | 34.1 |
| VPT-Deep Jia et al. (2022) | 78.8 | 90.8 | 65.8 | 98.0 | 88.3 | 78.1 | 49.6 | 81.8 | **96.1** | 83.4 | 68.4 | 68.5 | 60.0 | 46.5 | 72.8 | 73.6 | **47.9** | 32.9 | 37.8 |
| DA-VPT+ (ours) | **74.4** | **92.7** | **74.3** | **99.4** | **91.3** | **91.5** | **86.2** | **96.2** | 87.2 | **87.2** | **76.3** | **81.3** | **62.58** | **52.82** | **65.3** | **84.9** | 51 | **33.11** | **48.7** |

Table 7: Results of performance comparisons on the VTAB-1k benchmark with ViT-B/16 models pre-trained on ImageNet-21K.

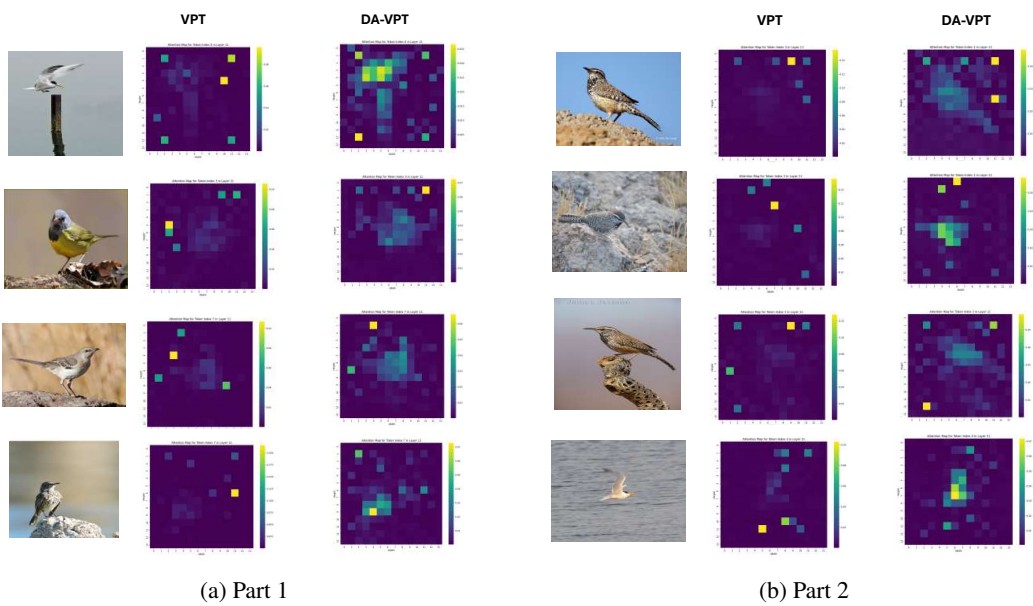

|(a) Part 1|(b) Part 2|

Figure 6: Here we show more visualization of the attention maps as examples. We randomly pick image from CUB and choose attention map according to the index of their assigned class.

Another limitation of our proposed method is the slightly higher latency compared to some other PEFT approaches. The introduction of the metric learning losses and the associated computations contribute to this increased latency. While the latency is still within acceptable limits for most practical applications, it is an aspect that could be further optimized. To address these limitations, our future work will focus on developing efficient strategies for hyperparameter search, such as employing advanced optimization techniques or meta-learning approaches. Additionally, we will explore ways to optimize the computational efficiency of our method, potentially through algorithmic improvements or hardware-specific optimizations, to reduce the overall latency without compromising the performance gains achieved by our approach.

Despite these limitations, our method demonstrates significant improvements in fine-tuning performance across various backbone models and downstream tasks, as evidenced by the experimental results. The benefits of our approach, such as improved accuracy and robustness, outweigh the current limitations, making it a valuable contribution to the field of parameter-efficient fine-tuning.

## E  BOARDER IMPACT

The proposed Distribution Aware Visual Prompt Tuning (DA-VPT) method has the potential to significantly impact the field of computer vision by improving the fine-tuning performance of ViT models. This can enhance the accuracy and efficiency of various downstream tasks, benefiting applications such as medical image analysis, wildlife conservation, and autonomous vehicles. The parameter efficiency of our method also enables the deployment of powerful computer vision models on resource-constrained devices, democratizing access to advanced visual recognition capabilities.

However, it is crucial to consider the potential negative consequences and ethical implications of our work, such as bias and fairness issues if the training data is not diverse and representative enough. Privacy concerns may also arise, particularly in applications involving personal data. To mitigate these risks, we encourage researchers and practitioners to prioritize fairness, transparency, and accountability in the development and deployment of models built upon our work.

In conclusion, our DA-VPT method has the potential to make a significant positive impact across various domains. However, it is essential to approach this technology with responsibility and care, ensuring that its benefits are realized while mitigating potential risks and negative consequences.

