# OpenReview forum: "Prompt Distribution Matters: Tuning Visual Prompt Through Semantic Metric Guidance"
_ICLR.cc/2025/Conference — ICLR 2025 Conference Withdrawn Submission_

### Official Review · Reviewer_TP9Z · 2024-10-30

**Soundness:** 2
**Presentation:** 3
**Contribution:** 2
**Rating:** 3
**Confidence:** 5

**Summary:**

This paper introduces a prompt-based PETL method, named Distribution Aware Visual Prompt Tuning (DA-VPT). DA-VPT explores the correlation between prompt tokens and image patch tokens by initializing the prompt parameters through clustering the image patch tokens and applying two metric learning-based loss functions to provide additional supervision signals on the prompt tokens. Experiments conducted on image classification tasks (VTAB-1k & FGVC) and semantic segmentation tasks (ADE20k & Pascal Context) demonstrate that DA-VPT outperforms the VPT baseline and other prompt-based methods. Extensive ablation studies validate the contributions of the key components within DA-VPT.

**Strengths:**

1. The overall writing is clear, and the proposed method is straightforward to follow.
2. The authors conduct experiments that effectively validate the proposed methods across various pre-trained models and downstream tasks.

**Weaknesses:**

1. The motivation for exploring the correlation between prompt tokens and image patch tokens closely aligns with the work presented in [a]. Additionally, the proposed dynamic mapping method appears to share a similar technical approach with [a]. The authors should provide a comparative analysis to clarify these connections.

2. After carefully checking the performance between DA-VPT and [a], it is observed that DA-VPT performs comparably to [a] when utilizing supervised-/MAE-/MoCov3-pre-trained ViT-B. For instance, DA-VPT achieves a mean accuracy of 83.20% on FGVC with the MAE pretrained ViT-B, while [a] achieves 83.26% under the same conditions. Given that DA-VPT is built upon [a], this raises concerns regarding the effectiveness of the metric-based loss function. The authors should conduct ablation studies using [a] as the baseline to isolate the impact of prompt initialization and clearly demonstrate the effectiveness of the proposed loss function, rather than using VPT as the baseline in Table 4.

3. It remains unclear whether the metric-based loss genuinely enhances the similarity between prompt tokens and image tokens of the same class, especially since [a] achieves this solely through initialization, as evidenced by the increased mutual information.

4. The manuscript lacks information regarding the number of prompt tokens and the hyper-parameters \beta and \alpha used across various datasets.  To improve reproducibility, the authors should provide a table to detailing the number of prompt tokens and hyper-parameters used for each dataset in the experiments.

5. Figure 1 (b) is confused. I found a lot of little squares with different colors, but I didn't understand how these squares convey the flow of information. The authors should provide a more detailed caption or legend to clarify what each color represents and how the arrangement of squares relates to the information flow.

[a] Revisiting the Power of Prompt for Visual Tuning. In ICML 2024.

**Questions:**

How many prompt tokens will be selected during salient patch selection? Furthermore, will substituting the attention map with the similarity between tokens for patch selection impact performance?

---

### Official Review · Reviewer_jQPx · 2024-10-31

**Soundness:** 3
**Presentation:** 2
**Contribution:** 2
**Rating:** 5
**Confidence:** 4

**Summary:**

This paper proposes a prompt tuning method with semantic metric learning guidance. The authors highlight the importance of distribution match between prompts and patch tokens, and realize the distribution match with metric learning techniques. The authors compare the proposed method with various PEFT methods on various benchmarks and backbones.

**Strengths:**

+ The authors have shown that the proposed method is effective with various backbones and benchmarks

+ The proposed method is reasonable.

**Weaknesses:**

- The major concern is the novelty of the motivation. [ref1] analyzed the distribution gap between prompts and patch tokens. The authors should include the detailed discussion with [ref1]. More specifically, [ref1] seems to have better performance than DA-VPT.

[ref1] Revisiting the Power of Prompt for Visual Tuning, ICML 2024

- Another concern is the fairness of evaluation. Compared to GateVPT and E2VPT, the proposed method has more learnable parameters. The authors should adjust the number of prompts to match the total number of parameters with baselines for a fair comparison.

- How can we guarantee that a prompt should capture the class information rather than indiscriminately distributed information (Lines 204--206)? ViT may work like, each prompt captures some low-level information and CLS token aggregate them to classify. This hypothesis should be verified either theoritically or empirically.

- Metric learning tend to be sensitive for the hyper-parameter values. I would recommend the authors to include the ablation study by varying the values of delta, beta, and gamma. And, I'd like to see the std of performance for each hyper-parameter setting

**Questions:**

1. What if the total number of classes is small value? For example, if the target classification task is binary classification, then the number of prompts should be just 2?

2. I need more clarification on Eq (4). What is P+?

---

### Official Review · Reviewer_TUiU · 2024-11-03

**Soundness:** 2
**Presentation:** 3
**Contribution:** 3
**Rating:** 6
**Confidence:** 5

**Summary:**

This paper mainly explores the relationship and distribution between prompts and image tokens. Specifically, in the architecture design, prompt learning is guided by distance metrics based on category-relevant visual information. The effectiveness of the method has been validated in recognition and segmentation tasks.

**Strengths:**

+ By conducting comprehensive comparative experiments, the superiority of the current architecture's performance has been effectively demonstrated.
+ The writing is logically clear with a well-defined motivation.

**Weaknesses:**

- There are equation inference errors and image citation errors, such as the inferences in lines 958-969 of the supplementary material and the citation in lines 292-294.
- Further experimental analysis and symbol meaning needs to be supplemented. For example, the strategy mentioned in lines 248-251 needs further elaboration in comparison to the performance differences in generating attention maps.

**Questions:**

1. The symbol representation in Equation 3 is not rigorous. K represents an individual class. The i and k on the left side of the equation should denote belonging to the same class. However, using two different letters to represent the same class can easily lead to ambiguity.
2. Further clarification is needed regarding the selection of the margin in Equation 3.
3. The symbol "W_Q^l" first appears in line 231 without a corresponding definition or explanation of its meaning.
4. Further experimental analysis needs to be supplemented. For example, the strategy mentioned in lines 248-251 needs further elaboration in comparison to the performance differences in generating attention maps.
5. The image referenced in lines 292-294 is incorrect and unrelated to the content described in the current paragraph.

---

### Official Review · Reviewer_v9Ak · 2024-11-03

**Soundness:** 2
**Presentation:** 2
**Contribution:** 2
**Rating:** 5
**Confidence:** 4

**Summary:**

This paper investigates the integration of metric learning and efficient fine-tuning to enhance visual prompt tuning for various downstream tasks. The authors propose a novel approach that utilizes proxy NCA loss and triplet loss to guide the tuning of visual prompts semantically. They also incorporate techniques like Visual Prompt Tuning (VPT) and bias tuning to achieve effective learning. The study aims to demonstrate the effectiveness of these methods in improving model performance, highlighting the importance of semantic guidance on the prompts.

**Strengths:**

+The approach of using proxies as visual prompts presents a novel view within the scope of prompt tuning.

+The method is validated across different tasks, demonstrating its strong generalization capabilities.

**Weaknesses:**

-While the idea of using proxies as prompts is commendable, there is a lack of additional innovation beyond this approach. Notably, the experimental section does not include comparisons with the original proxy NCA loss and triplet loss, which would strengthen the validation of the proposed method.

-Moreover, similar works, such as GLGait [1], have explored the combination of center loss and triplet loss, providing thought-provoking insights.

-Additionally, previous research has thoroughly investigated the integration of different efficient fine-tuning methods, such as GLoRA [2] and NOAH [3].

-The combination of Visual Prompt Tuning (VPT) and bias in this work seems to undermine the primary advantage of VPT, which is its ability to adapt to downstream tasks without modifying the backbone parameters.

- In summary, the innovations in this paper are limited, and the experimental comparisons are insufficient. For instance, existing methods, like Time- [4], have achieved notable performance of 78.4 on VTAB, demonstrating the competitive landscape in this area. Given the abundance of related prompt research, this paper does not provide enough insights or robust experimental results to warrant a recommendation.

[1].GLGait: A Global-Local Temporal Receptive Field Network for Gait Recognition in the Wild.
[2]. One-for-all: Generalized LoRA for parameter-efficient fine-tuning.
[3]. Neural Prompt Search.
[4]. Time-, Memory-, and Parameter-Efficient Visual Adaptation.

**Questions:**

Please refer to the weaknesses. In addition, there are two more questions as below:

1) Can the authors clarify the rationale behind combining VPT with bias.

2) In Tab 7, is KITTI/distance 65.3?

---

### Note · Authors · 2024-11-15

**Comment:**

Dear Reviewers,

We sincerely thank you for your detailed and constructive feedback. Although we have decided to withdraw our submission, we greatly appreciate the opportunity to address your major concerns and clarify certain aspects of our work.

1. Comparison with Wang et al. (2024):
We acknowledge that Wang et al.'s method, based on prompt initialization, reports performance comparable to ours. However, we did not include this work in our benchmarks for the following reasons:

- Reproducibility concerns: The authors of Wang et al. recently removed their code repository, which included discussions about reproducibility issues raised by others.
- Discrepancies in reproduced results: Even with their released code, our reproduction of their method still got results significantly below those reported in their paper.
- Effect on our framework: In our revised experiments, we found that their proposed prompt initialization method does not positively impact our framework's performance.

2. Key differences and advantages of our approach:

Our method fundamentally differs from prompt initialization by dynamically optimizing the distribution of prompts throughout the fine-tuning process. This allows us to adaptively control class assignments, leading to more effective learning.

3. Exploration of other metric learning losses and PEFT methods:

We carefully considered alternative metric learning losses (e.g., Proxy-NCA and Triplet Loss). However, these are less suitable for our framework due to the disparity between the number of visual prompts and data tokens. This imbalance often destabilizes training. In contrast, our use of the Proxy-Anchor loss explicitly accounts for the asymmetric nature of prompt-token relationships, ensuring stable optimization.

Lastly, we will address all other concerns raised in our next revision. Thank you once again for your valuable feedback, which will greatly inform the improvement of our work.

Sincerely,
Authors

**Withdrawal Confirmation:**

I have read and agree with the venue's withdrawal policy on behalf of myself and my co-authors.